# End-of-Life Care Mobile App for Intensive-Care Unit Nurses: A Quasi-Experimental Study

**DOI:** 10.3390/ijerph18031253

**Published:** 2021-01-30

**Authors:** Jin Hee Yang, Gisoo Shin

**Affiliations:** 1Intensive Care Uint, Seoul National University Hospital, 101 Daehak-ro, Jongno-gu, Seoul 03080, Korea; jininabottle@gmail.com; 2College of Nursing, Chung-Ang University, 84 Dongjak-gu, Heukseok-ro, Seoul 06974, Korea

**Keywords:** end-of-life care, intensive care unit, nurse, mobile app, efficacy

## Abstract

Intensive-care unit nurses may experience difficulties in end-of-life care because of frustration or lethargy. The purpose of this study was to develop a mobile end-of-life care program for intensive-care unit nurses and evaluate the effects on competence factors such as knowledge, self-efficacy, and compassion. A quasi-experimental design was used. The participants included 44 nurses who had less than three years of experience in the intensive-care unit, divided into the experimental group and control group. After the intervention, the experimental group showed a significant improvement in self-efficacy in end-of-life care and compassion in end-of-life care. Based on the results of this study, the end-of-life care mobile app was an effective educational method for nurses with experience of less than 3 years in an intensive-care unit. To improve the quality of end-of-life care, it is necessary to develop various educational programs considering the greater role of the fourth industrial revolution in the future.

## 1. Introduction

Most people hope to spend the final days of their life comfortably at home [1]. However, around 76.2% of the total deaths in Korea ended up occurring in hospitals, and the mortality rate in the intensive-care units (ICUs) is higher in Korea than in other Organization for Economic Cooperation and Development (OECD) countries [2]. As the death of most patients in an ICU is attributed to the discontinuation of life-sustaining treatment or sudden death during the course of treatment, ICU nurses who care for patients as well as their families could experience psychological distress [3]. Because of the nature of the ICU environment, critical situations often occur and the ICU nurses frequently experience burnout [4]. Thus, it is shown in South Korea that the staff turnover rate is high among ICU nurses [5]. Consequently, the nurses in ICUs are more frequently replaced and new recruits have less experience on average compared to other medical departments, in which they have 3 years of work experience or less [6]. Moreover, many of the newly recruited ICU nurses are recent nursing school graduates who just have become newly registered nurses and have little to no experience with end-of-life care [7]. Therefore, they may be unsure of how to cope with end-of-life care and what type of nursing care should be provided. In addition, they may have difficulties in maintaining emotional balance while providing end-of-life care and may experience frustration or lethargy [8].

End-of-life care is a nursing action that supports patients who are about to die and their families, and it refers to nursing in the final hours or days of patients. In this situation, nursing aims to help relieve the pain of patients with an untreated terminal illness and help them maintain their dignity and quality of life until the last moments of life [9]. Common nursing practices for end-of-life patients and their families include keeping them updated and preventing trauma from death through empathetic communication [10]. For these reasons, nurses providing end-of-life care in the ICU should have specific competence skills.

However, in ICUs in South Korea, the treatment of acute and severe patients is given the greatest attention, and there is a tendency to neglect end-of-life care [11]. As mentioned earlier, for nurses with limited ICU experience, they may lack overall knowledge of end-of-life care, end-of-life patient characteristics, or end-of-life procedures such as discontinuation of life-sustaining treatment [6]. In addition, they may lack compassion, including support for the families of patients who are about to die [9]. Compassion has been defined as a deep awareness of the suffering of another coupled with the wish to relieve it [12]. Moreover, self-efficacy is a crucial part of nursing competency to meet the requirements of end-of-life patients and their families. Self-efficacy in end-life-care is the belief that nurses who provide end-of-life care have the ability to fulfill the physical and psychological needs of patients who are dying [13].

With the emergence of many technological innovations, information technology has seen continuous evolution over the past years. The online exchanges of advice and experience between health professionals while discussing the digital records of clinical cases is one example [14]. The uses of mobile applications in particular have been applied to the various aspects of education including health-related programs [15]. According to the systematic research about mobile applications [16], the interventions based on mobile apps were found to be useful in improving various health practices due to the app’s ability to provide feedback or information with visualization. Mobile app-based educational intervention can also be delivered without the limitations of space or time at the learner’s convenience. It can even bridge the educational gaps between regions to certain extent by ubiquitously providing education everywhere including the areas where such intervention can be limited [17]. Thus, the intervention using a mobile application is often regarded to have high efficiency and efficacy [18]. With the ongoing global crisis of the coronavirus disease 2019 (COVID-19) pandemic, the fields of mobile education and telemedicine overall are projected to go through even faster development [14].

Despite such importance, the benefits of educational interventions using a mobile app have not been fully explored, particularly for the those aimed at ICU nurses. This study aimed to develop a mobile end-of-life care program and evaluate its efficacy as an app for ICU nurses. This study was focused on the following question: Overall, do ICU nurses who participate show changes in their end-of-life care knowledge, self-efficacy in end-of-life care, and compassion in end-of-life care following educational intervention using the mobile app?

## 2. Methods

### 2.1. Participants

This study involved nurses working in the adult ICU at Seoul National University Hospital, South Korea. For this study, we received the permission of the directors of the ICU and posted the public announcement on the hospital’s online staff portal from 29 August to 11 September 2019 to notify people that the study was recruiting ICU nurses with experience of 3 years or less. Those who wanted to participate could message back the researcher directly via the portal’s messenger feature. Once they contacted the researcher, they were instructed to attend the orientation session held by the researcher the day before the intervention’s start.

At the orientation, the participants were divided into the two groups of 24 to make up the experimental and control groups. After signing their consent forms, the ICU nurses of the experimental group installed the app on their phones after signing the consent form, and answered the participant questionnaire. After the invention was over, the data from 23 participants in the experimental and 21 in the control group were used for the final analysis, excluding data of the four nurses who withdrew during the study.

### 2.2. Ethical Considerations

This study was conducted with the approval of the Institutional Review Board of Seoul National University Hospital (No. 1909-136-975). After providing the study participants with information on the purpose of the study, the confidentiality of the data, and the disposal of the data after completing the study, voluntary written consent was obtained. The subjects could withdraw from the study any time during the study period and were told that there was no penalty for withdrawing. The collected research data were not used for any purpose other than for this research, and the data were discarded according to the Bioethics Act.

### 2.3. Measurements

We assessed the end-of-life care knowledge, self-efficacy in end-of-life care, and compassion in end-of-life care of the nurses. End-of-life care knowledge was assessed using the Korean Palliative Care Quiz for Nursing [19]. This tool consists of 20 items scored as 0 or 1 point; the higher the score, the higher the knowledge level. Self-efficacy in end-of-life care was measured using the revised vision of the Korean self-efficacy scale [20]. This scale consists of 14 items rated on a 4-point scale; the higher the score, the higher the self-efficacy in end-of-life care. Compassion in end-of-life care [21] consists of 13 items rated on a 5-point scale; the higher the score, the higher the compassion in end-of-life care.

### 2.4. Statistical Analysis

The *t*-test and chi-square test were used to analyze the effects and determine the homogeneity between the two groups. Differences in pre- and post-tests for each group were assessed using a paired *t*-test to determine the efficacy of the end-of-life care mobile app (EOL care app). Data are presented as the mean and standard deviation. All data analyses were conducted using SPSS 26.0 software (IBM, New York, USA). A *p*-value of <0.05 was considered to be statistically significant.

## 3. Procedure

### 3.1. Development of the End-of-Life Care Mobile App (EOL Care App)

We developed an EOL care app based on Android. The developed app consisted of an overview of the EOL care app and sections on pain management and managing the main symptoms of end-of-life patients, therapeutic communication skills, decision making in ethical conflict situations, and the experience of real end-of-life patients (Figure 1). For motivation and immersion, a picture of an actual machine used in the ICU was inserted along with other images, voice recordings, and videos. Moreover, nurses were able to self-reflect through a virtual deathbed experience based on real cases, and the surrogate experience, which provided participants with the nursing knowledge shared from the experienced ICU nurses, was also featured. The content validation index (CVI) of this program was 0.90 via five relevant experts including nursing school professors and the head nurses of the ICU.

### 3.2. Intervention in the Experimental and Control Groups

For subjects assigned to the experimental group, the EOL care app was downloaded and installed on the mobile phone before the intervention. In addition, an orientation was given on how to use the EOL care app, and nurses completed an anonymous online-based pre-test questionnaire. The pre-test questionnaire asked the nurses to create a unique identification code that would be used to identify their post-test responses. The experimental group accessed and learned about the app every day, and the average daily learning time was around 30 min for a total of seven days. The end-of-life care booklet was provided to the subjects assigned to the control group, and pre-intervention surveys were conducted for both the experimental and control groups (Figure 2). The contents of the end-of-life care booklet contained the educational instructions on life-sustaining treatment, pain management, and other relevant materials according to the guideline of Korean society for hospice and palliative care.

EOL care app intervention took place from 16 September to 4 October 2019, and a post-test questionnaire was conducted three days after the intervention ended. After intervention in the experimental group, the EOL app was also provided to the control group.

## 4. Results

### 4.1. General Characteristics and Homogeneity

The participants in this study included 34 females (79.1%) and 10 males (23.3%) with an average age of 26.14 years. Among them, 30 nurses (69.8%) had education experience in intensive care, and 11 nurses (25.6%) had experience in hospice palliative care.

Analysis of the homogeneity between the control and experimental groups showed that there were no significant differences (Table 1). Based on pre-test scores, there was no significant difference between the control and experimental groups in end-of-life care knowledge (*p* = 0.189), self-efficacy in end-of-life care (*p* = 0.448), and compassion in end-of-life care (*p* = 0.991).

### 4.2. End-of-Life Care Knowledge

The pre- and post-test scores of end-of-life care knowledge were compared (Table 2). In the experimental group, the total mean score of end-of-life care knowledge was increased from 0.44 ± 0.19 to 0.76 ± 0.14, and the difference was statistically significant. The total mean score of end-of-life care knowledge was also significantly improved in the control group (*p* < 0.001). There was no statistically significant difference in the pre- and post-test scores between the two groups (*p* = 0.086).

### 4.3. Self-Efficacy in End-of-Life Care

The change in the total mean score of self-efficacy in end-of-life care between pre- and post-tests was statistically significant in the experimental group (*p* ≤ 0.001). In addition, the difference in the pre- and post-test scores between the control and experimental groups was statistically significant (*p* ≤ 0.001) (Table 3).

### 4.4. Compassion in End-of-Life Care

The post-test total mean scores of compassion in end-of-life care in the control and experimental groups were 3.41 ± 0.68 and 4.08 ± 0.39 respectively. The score in the experimental group was increased by 0.73 ± 0.49, which was higher than that in the control group (0.08 ± 0.91), and the difference was statistically significant between the two groups (*p* = 0.005) (Table 4).

## 5. Discussion

The purpose of this study was to develop an EOL care app for ICU nurses in South Korea and examine the efficacy of this app. The findings of this study showed that the increase in the mean scores of self-efficacy in end-of-life care (0.23 vs. 1.01, *p* = <0.001) and compassion in end-of-life care (0.08 vs. 0.73, *p* = 0.005) was significantly different between the control and experimental groups. However, the increase in the mean score of end-of-life care knowledge was not significantly different between the control and experimental groups.

According to a study [22], around 71% of newly graduated nurses working in ICUs in South Korea were found to experience a patient’s death for the first time within less than three months of clinical work. The ICU nurses with experience of less than 3 years said that they lacked professional knowledge and had difficulty in providing the care required for the patient’s death. Consequently, we developed the EOL care app based on the various symptoms and signs of end-of-life patients to help nurses provide high-quality nursing care in various situations. In particular, the positive results of this study could be attributed to the development of the EOL app based on a manual of ICU procedures [23].

To improve the ICU nurses’ self-efficacy in end-of-life care in this study, a surrogate experience was included in the EOL app. The observation of the experience of others through the surrogate experience could increase the confidence of an individual [24]. Another study [25] showed that self-efficacy in end-of-life care was strengthened by providing information that senior nurses were able to collaborate effectively with medical staff or families.

Compassion is one of the main competence skills needed for end-of-life care [12]. End-of-life patients in an ICU are often dismissed as simple subjects of treatment surrounded by life-sustaining devices and without dignity as human beings [26]. ICU nurses should help alleviate the loneliness and anxiety of end-of-life patients through compassion, recognize them as holistic human beings, and help them and their families avoid experiencing death as a traumatic event [27]. Above all, it is necessary to provide end-of-life patients and their families with opportunities to participate in the nursing and decision-making process and communicate sincerely with them [10]. In this study, the EOL app provided the ICU nurses with the immersive stories from the real end-of-life patients and their families to provide an insight from their perspectives. By experiencing a virtual deathbed experience and learning about the will and feelings of colleagues who have experienced the death of a family member, nurses would be more prepared to manage trauma from death. As a result, compassion in end-of-life care was greatly improved in this study.

EOL care app intervention was possible without time and space limitations. In addition, it provided various difficulty levels according to the subjects’ skills or experience, and it was found to be effective in achieving the goals of participants with high satisfaction due to the high interest it generated. The positive effects of using the mobile app may be attributed to the ease of the ICU nurses in finding and accessing the desired content anytime without space and time limitations [17].

However, this study experienced limitations in several aspects. First, the limited range of the mobile application must be noted. The EOL care app in this study was designed for Android operating system and could not yet be implemented in iOS devices like iPhones. The study was also one-off intervention that targeted the limited population of the nurses in a single South Korean hospital over a short period. These limitations of the research on its software compatibilities and spatiotemporal scales must be considered and a follow-up study is needed to verify the EOL care app’s efficacy. Second, the app program was evaluated based on the nursing competency of the participants. While competency has been the idealized capacity of performing tasks in nursing, performance has been referred to as the occurrence of actual behaviors in numerous clinical situations [28]. Training to assess the competencies of nurses has been based on the performances in clinical settings. Because ICU nurses will eventually have to take care of actual end-of-life care patients, the performance of app users must also be assessed [29]. Therefore, further studies are needed to compare the competencies and performance of the participants throughout the uses of the EOL care app interventions.

## 6. Conclusion

In this study, we developed and evaluated an EOL care app to verify its efficacy in improving the nursing competence of nurses who had less than three years of end-of-life care experience in an ICU. The results showed that the self-efficacy in end-of-life care and compassion in end-of-life care of ICU nurses in this study were improved. This may be attributed to the effective acquisition of specific knowledge and skills in caring for end-of-life patients. The EOL care app provided an opportunity to examine the important role of nurses in supporting end-of-life patients and their families. The findings demonstrated the importance of developing end-of-life care programs based on various teaching–learning strategies. While future research should also address clearer evidences and long-term results that can objectively evaluate competency and performance, the present study demonstrated the potential for EOL care app-based education of ICU nurses to be widely utilized in future.

## Figures and Tables

**Figure 1 ijerph-18-01253-f001:**
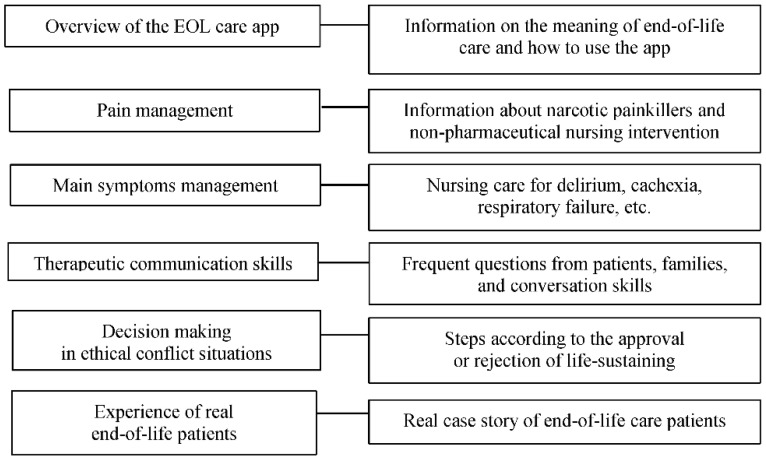
Contents of the end-of-life care application for intensive care unit nurses.

**Figure 2 ijerph-18-01253-f002:**
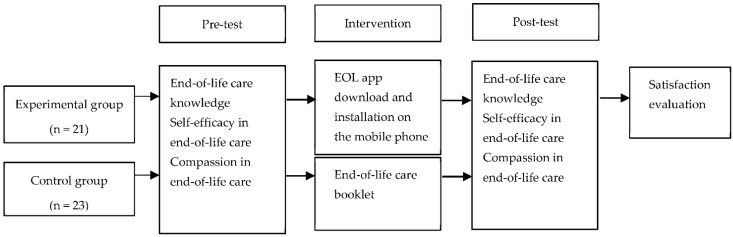
Flow diagram of the study. EOL app; end-of-life care mobile application.

**Table 1 ijerph-18-01253-t001:** General characteristics of the participants and homogeneity between the control group and experimental group.

	Control Group (n = 21)		Experimental Group (n = 23)		t/χ^2^	*p*
	N (%)	Mean ± SD	N (%)	Mean ± SD		
Mean age, years		25.14 ± 2.27		27.14 ± 2.52	2.71	0.501
Gender					0.14	1.000
Female	16 (76.2)		18 (78.3)			
Male	5 (23.8)		5 (21.7)			
Marital status					0.78	0.663
Single	19 (90.5)		20 (87.0)			
Married	2 (9.5)		3 (13.0)			
Religion					3.53	0.172
Yes	8 (38.1)		11 (47.8)			
No	13 (61.9)		12 (52.2)			
Education experience in intensive care					2.10	0.277
Yes	14 (66.7)		16 (69.6)			
No	7 (33.3)		7 (30.4)			
Education experience in hospice palliative care					0.00	1.000
Yes	5 (23.8)		6 (26.1)			
No	16 (76.2)		17 73.9)			
Count of end-of-life care per month		1.25 ± 1.03		1.16 ± 1.08	0.28	0.782
End-of-life care knowledge		0.45 ± 0.10		0.51 ± 0.93	−0.23	0.189
Self-efficacy in end-of-life care		2.46 ± 0.27		2.57 ± 0.24	0.77	0.448
Compassion in end-of-life care		3.33 ± 0.55		3.35 ± 0.32	−0.01	0.991

**Table 2 ijerph-18-01253-t002:** Difference in the pre- and post-test scores of end-of-life care knowledge between the experimental group and control group.

	Control Group (n = 21)			Experimental Group (n = 23)			Difference
	Mean ± SD		t(*p*)	Mean ± SD		t(*p*)	
	Pre	Post		Pre	Post		t(*p*)
End-of-life care knowledge	0.45 ± 0.10	0.59 ± 0.13	4.96 (<0.001)	0.44 ± 0.19	0.76 ± 0.14	6.86 (<0.001)	−1.76 (0.086)

**Table 3 ijerph-18-01253-t003:** Difference in the pre- and post-test scores of self-efficacy in end-of-life care knowledge between the experimental group and control group.

	Control Group (n = 21)			Experimental Group (n = 23)			Difference
	Mean ± SD		t(*p*)	Mean ± SD		t(*p*)	
	Pre	Post		Pre	Post		t(*p*)
Self-efficacy in end-of-life care	2.28 ± 0.25	2.52 ± 0.31	2.35 (0.050)	2.37 ± 0.29	3.37 ± 0.49	7.44 (<0.001)	−4.57 (<0.001)

**Table 4 ijerph-18-01253-t004:** Difference in the pre- and post-test scores of compassion in end-of-life care between the experimental group and control group.

	Control Group (n = 21)			Experimental Group (n = 23)			Difference
	Mean ± SD		t(*p*)	Mean ± SD		t(*p*)	
	Pre	Post		Pre	Post		t(*p*)
Compassion in end-of-life care	3.33 ± 0.55	3.41 ± 0.68	0.42 (0.676)	3.35 ± 0.32	4.08 ± 0.39	7.05 (<0.001)	−2.94 (0.005)

## Data Availability

The data presented in this study are available on request from the corresponding author.

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
