# Peer review of "End-of-Life Care Mobile App for Intensive-Care Unit Nurses: A Quasi-Experimental Study"

_ijerph, 2021, doi:10.3390/ijerph18031253_

Round 1

Reviewer 1 Report

Dear authors,

I hope to contribute constructively to your research with the following review.

The manuscript reports the validation process of a mobile application developed for use in training with nurses who specializes in the care of the patient at the end-of-life. The training enables professionals to interact with the work environment, collaborate effectively in the center of the doctors-family-patients triad, and maintain the balance of their emotions in the face of various scenarios. This set can provide psychological stability to professionals, optimizing their quality of personal and professional life. The optimization reflects directly on the target audience, considering humanizing the environments inserted in intensive care units.

Broad comments

The app is an active methodology tool with a focus on simulation. This approach can enhance the traditional teaching process and is an alternative for people who need a special learning regime.

This version of the manuscript presents the results clearly and objectively. The summary of Tables 2, 3, and 4 - together with their related topics - clarify what benefits the app incorporates to those involved. This new layout solves my criticisms of the previous work, in addition to intensifying the central idea of the manuscript.

This study is interdisciplinary, as it benefits the dialogue between doctors and nurses, the care for the sick and their families, and the occupational health of nurses, using a methodological computer science tool applied in education.

Sincerely,

Reviewer.

Author Response

We really appreciate your comment.

Your comments will gear up our effort to conduct more significant studies in the future.

Reviewer 2 Report

Accept

No further comments

Author Response

We really appreciate your review.

Your review will gear up our effort to conduct more significant studies in the future.

Reviewer 3 Report

Introduction provides sufficient details and it is focused on introducing the background of the aim of the paper. Please consider citing a recently published article on MDPI that evaluates the use of mobile apps and telemedicine to assist patients and operators (doi: 10.3390/jcm9061891). It could help to broaden your introduction

Methods are well-organized and accurately described

Results appear clearly presented and undestandable

Discussion and conclusion are supported by results and provide adequate context with the existing literature 

Author Response

We are honored to view and refer to your incredible study, the literature review about teleorthodontics between dentist and patient of demonstrating the benefits and limitations of the technological innovations. We have referred to this study as you have advised.

Revision:

With the emergence of many technological innovations, information technology has seen the continuous evolution over the past years. The online exchanges of advices and experiences between health professionals while discussing the digital records of clinical cases is one example [14]. The uses of mobile applications especially have been applied to the various educations including health-related programs [15]. According to the systematic research about mobile applications [16], the interventions based on mobile apps were found to be useful on improving various health practices due to the app’s ability to provide feedback or information with visualization. Mobile app-based educational intervention can also be delivered without the limitations of space or time at learner’s convenience. It can even bridge the educational gaps between regions to certain extent by ubiquitously providing educations everywhere including the areas where such intervention can be limited [17]. Thus, the intervention using a mobile application is often regarded to have high efficiency and efficacy [18]. With the ongoing global crisis of the COVID-19 pandemic, the fields of mobile educations and telemedicine in overall are projected to go through even faster developments [14].    

Despite such importance, the benefits of educational interventions using a mobile app has not been fully explored, particularly for the one aimed at ICU nurses. This study aimed to develop a mobile end-of-life care program and evaluate its efficacy as an app for ICU nurses. This study was focused on the following question: Overall, do ICU nurses who participate show changes in their end-of-life care knowledge, self-efficacy in end-of-life care, and compassion in end-of-life care following educational intervention using the mobile app?

Reviewer 4 Report

This is a simple technical report, I do not think it is appropriate for publication in this journal. It is good to do this kind of applications, but the publication require more strict structure for the paper and its presentation. For example, the paper should include literature review, method comparison, innovation summary, etc. Relevant experiments and contents should be supplemented well, and the paper should be rewritten well.

Author Response

We really appreciate your comment. Based on the suggestions from the reviewers, the further details have been added to the introduction, discussion, and conclusion to strengthen the structure and presentation of the manuscript.

This study was the validation process of a mobile application developed for training with the nurses who specialize in the care of the end-of-life patients. Beyond the technical aspects, we believe that the study of the app can enhance the nurses in their competence to interact with work environment, collaborate effectively as the center of doctor-family-patient triad, and maintain their emotional balance in various scenarios.

This manuscript is a resubmission of an earlier submission. The following is a list of the peer review reports and author responses from that submission.

Round 1

Reviewer 1 Report

Dear authors,

I hope to contribute constructively to your research with the following review.

The manuscript reports the validation process of a mobile application developed for use in training with nurses who specializes in the care of the patient at the end-of-life. This activity enables better interaction between professionals with their work environment, collaborate effectively in the doctors-family-nurses triad, and maintain the balance of their emotions in the face of various scenarios. This set helps in the psychological stability of professionals, optimizing their quality of personal and professional life.

Broad comments

The app is an active methodology tool with a focus on simulation. This approach can enhance the traditional teaching process and is an alternative for people who need a special learning regime.

Inserting specific quotes from tables 2, 3, and 4 in the discussion facilitates the reader to associate the relationship between the resource used in the app and the benefit described in the tables. I advise doing briefly during the discussion.

This study is interdisciplinary, as it benefits the dialogue between doctors and nurses, the care for the sick and their families, and the occupational health of nurses, using a methodological computer science tool applied in education.

Specific comments

All keywords must be contained in the abstract.

Sincerely,

Reviewer.

Author Response

Response to the comments of Reviewer on ijerph ID:939804

Authors Response to Comments / Changes made to article

Page

I hope to contribute constructively to your research with the following review.

The manuscript reports the validation process of a mobile application developed for use in training with nurses who specializes in the care of the patient at the end-of-life. This activity enables better interaction between professionals with their work environment, collaborate effectively in the doctors-family-nurses triad, and maintain the balance of their emotions in the face of various scenarios. This set helps in the psychological stability of professionals, optimizing their quality of personal and professional life.

We really appreciate your comment.

Your helpful advises will gear up our effort to conduct more significant studies in the future.  

We have reflected your suggestions in the manuscript. Please refer to the revised texts highlighted in blue.  

Broad comments

The app is an active methodology tool with a focus on simulation. This approach can enhance the traditional teaching process and is an alternative for people who need a special learning regime.

Inserting specific quotes from tables 2, 3, and 4 in the discussion facilitates the reader to associate the relationship between the resource used in the app and the benefit described in the tables. I advise doing briefly during the discussion.

We now have summarized the significant results from the tables in the beginning of the discussion.

p6

This study is interdisciplinary, as it benefits the dialogue between doctors and nurses, the care for the sick and their families, and the occupational health of nurses, using a methodological computer science tool applied in education.

Specific comments

All keywords must be contained in the abstract

Sincerely,

Reviewer.

Thank you for your constructive comments. As you have suggested, we have revised the abstract.

Regards,

Authors

p1

Reviewer 2 Report

Thank you for an interesting manuscript on a important subject. 

The manuscript needs a major revision based on the following recommendations:

  1. The language should be improved by proof-reading of the whole manuscript.
  2. The introduction is too long and the aim should be described more clear and in more detail.
  3. I do miss a subheading about Ethical considerations and ethics approval under point 2. 
  4. The development of the mobile app is unclear. Please describe in more detail.
  5. Table 2, and 4 are too big and difficult to read. They should be adapted to be more readable and easier to understand for the reader.
  6. The discussion should start with a summary of the main findings. The effects of using the app should be discussed in more detail.
  7. A section with limitations is needed under an own subheading. There should be a more critical reflection of the methodology used.

Author Response

Response to the comments of Reviewer on ijerph ID:939804

Authors Response to Comments / Changes made to article

Page

Thank you for an interesting manuscript on a important subject. 

The manuscript needs a major revision based on the following recommendations:

We really appreciate your comment.

Your constructive comments will gear up our effort to conduct more significant studies in the future.  

We have reflected your suggestions in the manuscript. Please refer to the revised texts highlighted in blue.

The language should be improved by proof-reading of the whole manuscript.

Thank you for your advice. Because writing a paper in English is always challenging to non-English speakers like us, we really appreciate such comment.

As you have instructed, we have sought and adapted the proofreading advises from the American editor working at the professional editing agency Essayreview.

The introduction is too long and the aim should be described more clear and in more detail.

As you have mentioned, we have rewrote the introduction in more concise manners and described the aim of the study with more clear details.

p2

I do miss a subheading about Ethical considerations and ethics approval under point 2. 

Following your instruction, we have included the ethical considerations and ethics approval under point 2.

p2

The development of the mobile app is unclear. Please describe in more detail.

To address your advice, we have more thoroughly described the development of the app in the section 3.1.

p3

Table 2, and 4 are too big and difficult to read. They should be adapted to be more readable and easier to understand for the reader.

We agree and revised the tables to reflect your suggestions.

Pp5-6

The discussion should start with a summary of the main findings. The effects of using the app should be discussed in more detail.

We appreciate your good comments.

As you have suggested, we have revised the discussion section to include the summary of the study results.

Pp6-7

A section with limitations is needed under an own subheading. There should be a more critical reflection of the methodology used.

Thank you for your advice. However, to follow the author’s guideline of IJERPH, we could not place the limitation section under its own subheading. We instead have further elaborated the limitation section to include the effectiveness according to gender-based differences in the EOL care app

p7

Reviewer 3 Report

Thank you for the opportunity to review this manuscript. I offer the following specific evaluative comments:

The manuscript requires extensive editing with respect to grammar and clarity of writing. While the writing is clearer in the presentation of statistical data related to findings the rest of the paper was more challenging to navigate.

The issue of providing end of life care in the intensive care unit is an important topic. That nurses in the ICU may feel challenges to provide care that embraces a palliative care philosophy has been previously identified in the literature, for the reasons cited by the authors in the statement of the problem. Educational interventions that do not involve an app have been developed and evaluated. The authors provide rationale for the need to develop an app based on the grounds that information nurses need to support them in their work can be more readily accessed, compared to previous approaches to education. 

Suggestions for improving the paper include:

Articulation of specific research questions/hypotheses driving the study. It seems to me that this project might best be disseminated by the crafting of two separate manuscripts. One would deal more directly and deeply with the development and testing of the app as the intervention; the second would deal with the evaluation of the intervention.

Justification for the choice of a quasi-experimental design versus another design would be helpful.

Justification for recruiting newly graduated nurses, and those with minimal experience to be the sample is provided; however, one could also argue that even nurses who have worked for a long period of time in the ICU may also have some of the same struggles with providing end of life care, particularly given the ethos of the ICU, and the fact that they too would not have received any kind of systematic training in palliative care as part of their educational preparation. It is not clear what the authors mean in referring to being reprimanded by senior nurses. This needs to be specified.

As regards the sample, it is not clear what is meant by the statement, "...to obtain a total of 42 nurses patients, ..." Patients are not part of the study.

More information is required as to the processes used to recruit the nurses. That they were recruited is known, but how did they gain access to the nurses? Did the authors make personal presentations to the nurses on various shifts? Were they contacted by email? 

The instrumentation used for the study appears to be appropriate given the outcome variables of interest. While the authors report on the psychometric performance of the tools in their study, it would have been helpful to have included their general psychometric properties as reported in the literature. Was evaluation of tool performance also a stated aim of the study? if so, it should be included as part of the study purpose.

The construct of compassion has been quite extensively investigated, and an empirical model of its salient constituents, as well as health care provider and patient perspectives of how it might be taught/learned. See citations from author Dr. Shane Sinclair in PubMed. Integration of this literature would strengthen the manuscript. The authors are are cautioned not to conflate empathy and compassion, as in the discussion the two terms are used seemingly interchangeably. There are different.

The primary limitation discussed about the study is the limitation of the technology (i.e. not operable on iPhone). What other limitations are there, in light of the design that was used? Might there be some limitation in view of the fact that the majority of participants were female, given we know that there are attitudinal differences between men and women regarding death anxiety?

I am not familiar with nurse's t test. Did the authors mean the student's t-test?

Were the pre and post surveys mailed to participants and returned by mail to the investigators? Were any reminder notices sent? 

How were threats to validity managed in the study?

Author Response

Response to the comments of Reviewer on ijerph ID:939804

Authors Response to Comments / Changes made to article

Page

Thank you for the opportunity to review this manuscript. I offer the following specific evaluative comments:

The manuscript requires extensive editing with respect to grammar and clarity of writing. While the writing is clearer in the presentation of statistical data related to findings the rest of the paper was more challenging to navigate.

The issue of providing end of life care in the intensive care unit is an important topic. That nurses in the ICU may feel challenges to provide care that embraces a palliative care philosophy has been previously identified in the literature, for the reasons cited by the authors in the statement of the problem. Educational interventions that do not involve an app have been developed and evaluated. The authors provide rationale for the need to develop an app based on the grounds that information nurses need to support them in their work can be more readily accessed, compared to previous approaches to education. 

We really appreciate your comment.

Your useful suggestions will gear up our effort to conduct more significant studies in the future. 

As you have instructed, we have consulted and followed the proofreading advises from the American editor working at the professional editing agency called Essayreview.

We have reflected your suggestions in the manuscript. Please see the revised texts highlighted in blue.

Suggestions for improving the paper include:

Articulation of specific research questions/hypotheses driving the study. It seems to me that this project might best be disseminated by the crafting of two separate manuscripts. One would deal more directly and deeply with the development and testing of the app as the intervention; the second would deal with the evaluation of the intervention.

We agree with your suggestion.

However, since our main goal for the study was to evaluate the changes in the knowledge, self-efficacy, and compassion in the end-of-life care following the mobile education intervention, we have decided to focus on these topics through a single study this time.  

We instead have revised the contents from page 1 to 3 to articulate the research question more specifically.

p1, 3

Justification for the choice of a quasi-experimental design versus another design would be helpful.

Quansi-experimental design is generally used when RCT approach is difficult in the studies of medicine, nursing, or education. We have used the following design to validate the efficiency of the developed application.

p1-2

Justification for recruiting newly graduated nurses, and those with minimal experience to be the sample is provided; however, one could also argue that even nurses who have worked for a long period of time in the ICU may also have some of the same struggles with providing end of life care, particularly given the ethos of the ICU, and the fact that they too would not have received any kind of systematic training in palliative care as part of their educational preparation. It is not clear what the authors mean in referring to being reprimanded by senior nurses. This needs to be specified.

We agree with your suggestions about how the experienced nurses in the ICU can have similar struggles.

For the discussion about being reprimanded by senior nurses, we thought such discussion was too irrelevant to the study’s main topic, so we have decided to remove it.

p2

As regards the sample, it is not clear what is meant by the statement, "...to obtain a total of 42 nurses patients, ..." Patients are not part of the study.

Thank you for the helpful findings. We have corrected the typo to “…a total of 42 nurses,…”

p2

More information is required as to the processes used to recruit the nurses. That they were recruited is known, but how did they gain access to the nurses? Did the authors make personal presentations to the nurses on various shifts? Were they contacted by email? 

Following your instruction, we have provided the detailed explanation of the processes on page 2 and page 4. Please refer to the mentioned pages.

p2. 4

The instrumentation used for the study appears to be appropriate given the outcome variables of interest. While the authors report on the psychometric performance of the tools in their study, it would have been helpful to have included their general psychometric properties as reported in the literature. Was evaluation of tool performance also a stated aim of the study? if so, it should be included as part of the study purpose.

The primary purpose of the following study was to use the instrumentation to evaluate the effectiveness of the mobile end-of-life care, but not the performance of the instrument itself. Thus, we did not include it as the purpose of the study.

p2

The construct of compassion has been quite extensively investigated, and an empirical model of its salient constituents, as well as health care provider and patient perspectives of how it might be taught/learned. See citations from author Dr. Shane Sinclair in PubMed. Integration of this literature would strengthen the manuscript. The authors are cautioned not to conflate empathy and compassion, as in the discussion the two terms are used seemingly interchangeably. There are different.

Thank you so much for your advice. As you have recommended, we have edited the discussion about compassion and added the reference from Dr. Shane Sinclair.

p7, 9

The primary limitation discussed about the study is the limitation of the technology (i.e. not operable on iPhone). What other limitations are there, in light of the design that was used? Might there be some limitation in view of the fact that the majority of participants were female, given we know that there are attitudinal differences between men and women regarding death anxiety?

We now have further discussed about the limitation to include the effectiveness according to gender-based differences in the EOL care app.

p7

I am not familiar with nurse's t test. Did the authors mean the student's t-test?

Thank you for finding out the mistake. We have corrected the typo to nurse’s t-test.

p3

Were the pre and post surveys mailed to participants and returned by mail to the investigators? Were any reminder notices sent? 

As mentioned earlier, more detailed explanations of the processes is now described on page 2 and page 4. Please refer to those pages

p2, 4

How were threats to validity managed in the study?

We have revised the validity management in more details.

p3

Round 2

Reviewer 2 Report

Thank you for the revision and addressing my comments. I have no further comments. 

Author Response

Response to the comments of Reviewer on ijerph ID:939804

Authors Response to Comments / Changes made to article

Page

Thank you for the revision and addressing my comments. I have no further comments.

We want to express our appreciation for all your helpful comments. Your advice will help us to conduct more significant studies in the future
